# Neurobiology of Propofol Addiction and Supportive Evidence: What Is the New Development?

**DOI:** 10.3390/brainsci8020036

**Published:** 2018-02-22

**Authors:** Ming Xiong, Nimisha Shiwalkar, Kavya Reddy, Peter Shin, Alex Bekker

**Affiliations:** Department of Anesthesiology, New Jersey Medical School, Rutgers University, Newark, NJ 07107, USA; dr.nimisha4u@gmail.com (N.S.); kn268@njms.rutgers.edu (K.R.); shinpy@njms.rutgers.edu (P.S.); bekkeray@njms.rutgers.edu (A.B.)

**Keywords:** propofol, abuse potential, GABA_A_ receptor, glycine receptor, fospropofol

## Abstract

Propofol is a short-acting intravenous anesthetic agent suitable for induction and maintenance of general anesthesia as well as for procedural and intensive care unit sedation. As such it has become an unparalleled anesthetic agent of choice in many institutional and office practices. However, in addition to its idealistic properties as an anesthetic agent, there is accumulating evidence suggesting its potential for abuse. Clinical and experimental evidence has revealed that not only does propofol have the potential to be abused, but also that addiction to propofol shows a high mortality rate. Based on this evidence, different researchers have shown interest in determining the probability of propofol to be an addictive agent by comparing it with other drugs of abuse and depicting a functional similitude that involves the mesocorticolimbic pathway of addiction. In light of this, the Drug Enforcement Agency and the American Society of Anesthesiologists have put forth certain safety recommendations for the use of propofol. Despite this, the abuse potential of propofol has been challenged at different levels and therefore the preeminent focus will be to further validate the linkage from medicinal and occasional use of propofol to its addiction, as well as to explore the cellular and molecular targets involved in establishing this linkage, so as to curb the harm arising out of it. This review incorporates the clinical and biomolecular evidence supporting the abuse potential of propofol and brings forth the promising targets and the foreseeable mechanism causing the propofol addiction phenotypes, which can be called upon for future developments in this field.

## 1. Introduction

Propofol is a sedative hypnotic used widely in different healthcare settings such as operating rooms and offices. Though it is currently not classified as a controlled substance, there is increasing evidence that propofol has abuse potential and presents risks of addiction that should warrant a re-consideration of this regulation. In this review, we will discuss the evidence supporting increasing incidence of propofol abuse and the comparability of its actions with other substances of addiction such as alcohol and nicotine, in particular on the mesolimbic reward system. 

## 2. Clinical Properties of Propofol

Propofol is a fast-acting intravenous hypnotic medication extensively used for induction and maintenance of general anesthesia, monitored anesthesia care (MAC), and for conscious sedation in the setting of intensive care units (ICUs) of hospitals. Additional uses for propofol include refractory agitated delirium and as an antiemetic in patients with end-stage diseases or patients undergoing chemotherapy [1]. Propofol may also be used for the treatment of intractable status epilepticus, delirium tremens, and status asthmaticus and has anti-inflammatory properties that may be useful in sepsis and traumatic brain injury. Propofol has several advantages over other sedatives due to its rapid onset, short duration of action, faster metabolism, and rapid clearance. It has minimal or no residual side effects and hence it is widely used for anesthesia and sedation.

Propofol’s principal mechanism of action is to enhance chloride current at the gamma-aminobutyric acid type A (GABA_A_) receptor on the post-synaptic membrane of neurons. This mechanism is also responsible for its antiepileptic and anxiolytic effects [2]. Glycine receptors and ligand-gated chloride channels, which mediate neuronal inhibition at the spinal level, have also shown sensitivity to propofol [3].

## 3. Abuse Potential of Propofol

Although the vast majority of reported abuse of propofol has been by medical professionals, risk of its abuse in the general population has been highlighted in numerous case reports and case studies worldwide time and again [4,5]. Propofol’s incrimination in pop music superstar Michael Jackson’s death in 2009 resulted in giving more momentum to the awareness regarding propofol addiction. However, abuse of propofol has been widely reported since 1992. Propofol induces psychotropic effects similar to other drugs of addiction and clinical research involving healthy volunteers showed that sub-anesthetic doses of propofol induced feelings of “drunk”, “dizzy”, “elated”, and “high”, along with decreased feelings of “in control of thoughts” and “in control of body” [6]. A random-choice trial was conducted in healthy volunteers in which subjects were asked to choose between propofol and placebo (Intralipid, a soy-based lipid emulsion) after blind bolus injections of either propofol or placebo. This trial resulted in a statistically significant difference in choosing propofol over placebo than would be expected by chance. Moreover, propofol was chosen for its pleasant subjective effects whereas the placebo was chosen for unpleasant effects associated with propofol administration such as dizziness, confusion, or weakness. These trials show that propofol, as with other drugs of abuse, can induce pleasant feelings that can increase its risk for recreational use as well as addiction.

Studies have shown an increase in the reporting of drug abuse that can be attributed to propofol. An analysis of 22 cases of propofol addiction in healthcare providers treated at a large addiction center between 1990 and 2010 was done by Earley et al. [7] who found that there was 25% increase in admissions reporting propofol use, in each semi-decade, in the treatment groups. Anesthesia providers constituted a majority of the cases reported, and all providers had relatively easy access to propofol. Self-reported reasons for propofol use included insomnia, anxiety, and seeking euphoria. Noteworthy is the possibility of second-hand exposure to propofol through air in the operating room which produces sensitization to the reinforcing effects of propofol, thereby possibly driving towards addiction with repeated exposure; however, the evidence to prove it is very miniscule [8,9]. In an e-mail survey of 126 academic anesthesiology training programs in the US in 2006, observed incidence of propofol abuse over the past 10 years was 10 per 10,000 anesthesia providers with a high incidence of mortality (28%) among those abusing propofol [10]. This represents an increased incidence of reported propofol abuse compared to a prior survey from 1990 to 1997 which had a calculated 10-year incidence rate of 0.02%. The survey also found that there was a significant association between error in propofol accounting and episodes of abuse. In a retrospective survey of substance abuse in anesthetists in Australia and New Zealand from 2004 to 2013, propofol was the most commonly abused substance (41%) [11]. Propofol was inculpated in all eight cases that resulted in death, in which three were identified as suicides and five as overdose, highlighting the significant mortality associated with propofol abuse. These studies demonstrate a trend towards an increasing incidence of propofol abuse and the significant mortality associated with its abuse.

## 4. Status of Propofol as a Controlled Drug

The Drug Enforcement Agency (DEA) issued its final rule effective 5 November 2009 placing fospropofol, a prodrug of propofol, into schedule IV of the Controlled Substances Act [12]. Fospropofol’s increased bioavailability when taken orally versus propofol’s negligible oral bioavailability increases its risk for abuse [13]. Thus, the regulatory controls and criminal sanctions of schedule IV are applicable to the manufacture, distribution, dispensing, importation, and exportation of fospropofol and products containing fospropofol. Accordingly, fospropofol will remain readily available for use by responsible clinicians in urgent and emergent situations but will require that it be reasonably stored in a restricted and secure environment with access limited to qualified clinicians. On the other hand, though the DEA proposed its inclusion into schedule IV of the Controlled Substances Act in October 2010, propofol, which is the active metabolite of fospropofol, is not yet on the list of controlled substances [14]. This proposal has been supported by the American Society of Anesthesiologist (ASA), which unequivocally maintains that propofol should be used in a medical setting by professionals trained in provision of general anesthesia and with proper supervision by a physician trained in anesthesia and qualified to provide rescue, should it be required [15]. American Association of Nurse Anesthetists (AANA) recommends that propofol be placed in a secure environment only accessible by the professionals identified in a medication management policy thereby taking measures towards decreasing drug diversion [16].

Others have argued against classifying propofol as a controlled substance. Some properties of propofol including pain on injection and rapid onset have been cited against its potential for abuse. Due to propofol’s advantages over other sedatives, it has become an indispensable tool in modern anesthesiology practices. Impeding access to such a ubiquitous medication may unnecessarily add administrative load and cost to implement controls and delay access during critical emergencies. Therefore, any such action should be viewed with respect to these forthcoming challenges.

## 5. Molecular and Cellular Mechanisms of Propofol Addiction

The abuse potential of propofol that is observed clinically can be very well substantiated through animal studies [17]. Gatch et al. [18] showed that propofol produces a discriminable stimulus similar to known drugs of abuse. There is a growing evidence showing that propofol can be self-administered [17,19] and induces a conditioned place preference [20,21], which are the classic methods to evaluate the abuse potential of drugs. This implies that propofol, as with other drugs of abuse, is subject to reinforcing effects. The abuse potential of propofol has been further studied by various labs at the neurocircuitry, molecular and cellular levels and is discussed in this section. 

The mesocorticolimbic pathway is the key component of reward circuitry involved in addiction to drugs [22,23,24]. Evidence shows that different drugs of abuse exert their action on different primary target proteins to bring about behavioral and physiological rewarding effects of the drug. However, with acute and eventually chronic use, the drugs which have a potential for abuse converge on a common pathway of addiction which involves, most commonly, the mesolimbic dopaminergic output of Ventral Tegmental Area (VTA) to the Nucleus Accumbens (NAc) [25]. Most drugs of abuse encode for a novel or unexpected reward and result in large but transient increases in NAc dopamine levels via phasic firing of VTA dopamine neurons. Similar increase also results from specific cues related to the drug. The thus increased dopamine mediates the motivation to procure the reinforcer via low affinity dopamine D1 receptors acting on the direct striatal pathway of reward as well as sustain this motivation via high affinity dopamine D2 receptors inhibiting the indirect striatal pathway of aversion. In due course, synaptic and neuronal plasticity develops and results in craving for the drug even with lower increases in dopamine levels observed in chronic drug abusers [26].

The NAc is closely connected to emotion-regulating areas of the brain which are modulated by dopamine to motivate goal-directed behaviors in response to novel stimuli. Dopamine influences the forming of these responses via downstream transcription and protein synthesis, the products of which are targets of study for drugs of addiction. Animal model studies have been instrumental in studying these drug-induced changes. Several studies have been performed to compare the effect of propofol with other drugs of abuse, on the mesolimbic reward circuit. A study by Pain et al. [27] used micro dialysis to measure dopamine levels in the NAc of rats after different doses of propofol. Propofol was injected intraperitoneally at the following doses: 0, 9, 60, 100 mg/kg. Dopamine concentrations were decreased at 9 mg/kg dose but were largely increased by approximately 90% in both sub-anesthetic dose of 60 mg/kg and at anesthetic dose of 100 mg/kg, with anesthetic doses resulting in prolonged increase in dopamine levels. However, their results cannot differentiate between the increased release of dopamine or inhibition of dopamine re-uptake that resulted in increased dopamine levels.

A study by Wang et al. [19] focused on the significance of extracellular signal-regulated kinase (ERK) signal transduction pathways in NAc during propofol self-administration. The study showed that there is significantly increased expression of phosphorylated EKR (p-ERK) in the NAc in rats maintained on propofol, and pretreatment with SCH23390 (a D1 receptor antagonist) inhibited propofol self-administration and diminished the expression of p-ERK in the NAc. Intra NAc injection of ERK kinase inhibitor, U0126 (4 µg/side), attenuated the propofol self-administration. These results were consistent with several previous studies showing an increased p-ERK expression in the NAc following cocaine and ethanol self-administration [28]. Thus, the study hypothesized that ERK signal transduction pathways coupled with D1 dopamine receptors in the NAc is critical for downstream regulation of DA signal transmission in the NAc, mediating the increased self-administration and rewarding effects observed with propofol.

The Finkel-Biskins-Jinkins murine osteosarcoma viral oncogene homolog B (FosB) is a transcription factor induced by drugs of addiction [29]. The FosB delta variant (ΔFosB) is a truncated form of FosB with high stability, which accumulates with repeated exposure to drugs of addiction [30]. FosB and Fos family of proteins are induced rapidly and transiently in specific brain regions by drugs of addiction, most notably in the NAc and dorsal striatum. However, with chronic exposure to drugs of addiction, modified isoforms of ΔFosB accumulate in certain brain regions for several weeks after drug cessation, due to their long half-lives and their resistance to tolerance as compared to other FosB family proteins [31]. Induction of ΔFosB in the dynorphin-positive medium spiny neurons of the NAc has been shown to cause increased sensitization to the drug effects as well as resulting in incentive salience and predisposition to withdrawal symptoms and relapse [31,32]. In our lab, M. Xiong et al. [33] found an upregulation of ΔFosB protein in NAc with propofol administration, comparable with levels found after ethanol and nicotine administration. Drugs were administered via intraperitoneal injection twice a day for 7 days and NAc levels of ΔFosB were compared through protein and mRNA measurements via Western blot analysis and quantitative real-time polymerase chain reaction, respectively. There was also an upregulation of dopamine receptor D1 (drD1) which is an upstream signal molecule of ΔFosB within the addictive mesolimbic circuitry. 

K-Y. Li et al. [34] studied the effects of nanomolar (nM) concentrations of propofol on VTA neurons, which coincides with the minute concentrations present in the air of operating rooms, thus emphasizing on the involvement of second-hand exposure to propofol. It was observed that 1 nM propofol increased the frequency but not the amplitude of spontaneous excitatory post-synaptic current (EPSC) while increasing the amplitude but not the paired pulse ratio of evoked EPSC in VTA dopamine neurons, all of which depict a presynaptic mechanism. This effect was blocked by drD1 antagonist. In a nutshell, it was inferred that nanomolar concentrations of propofol, via dopamine D1 receptors in the glutamatergic terminals on to VTA neurons, can increase excitability of VTA dopaminergic neurons. However, when M. Xiong et al. [35] extended this observation to examine the levels of propofol in anesthesiologists exposed to expired gases of patients undergoing propofol-based intravenous sedation, we did not find any significant difference. However, this study used a cut-off value of 50 ng/mL and hence the possibility of lower levels of blood propofol cannot be excluded. Also, this study being performed in a well-ventilated operating room cannot be applied to remote location procedure rooms which perform the majority of propofol sedation cases.

Dopamine and cyclic-adenosine monophosphate (c-AMP)-regulated phosphoprotein of apparent Mr 32000 (DARPP-32) is positioned to play a key factor in either mediating or modulating the short-, and perhaps long-term effect of drug of abuse such as nicotine, ethanol, cocaine or amphetamines [36]. A study by Pavkovic et al. [37] looked at the effect of propofol administration on several target molecules which are also affected by other addictive drugs. Western blot and immunohistochemistry was performed on target brain regions namely the medial prefrontal cortex, the striatum, and the thalamus, 24 to 48 h after propofol anesthesia. Molecular targets that were examined included drD1 expression; dopamine release; FosB and pDARPP-32 positive cells; and activity of calcium/calmodulin-dependent protein kinase II-α (CaMKII α), a biochemical sensor of synaptic activity, which in turn causes behavioral sensitization [38,39]. These studies found significant increase in the CaMKII α in the prefrontal cortex and striatum 24 h after exposure to propofol anesthesia. There was also significant increase in the number of FosB and pDARPP-32 positive cells in the paraventricular nucleus of the thalamus (PVT), which is consistent with studies performed in other drugs of addiction.

Some research groups also present the plausible role of other neurotransmitters and receptor substrates in promoting propofol addiction. For example, working along the lines of previous experiments which have linked addiction with glucocorticoid receptor signaling [40,41], some of the recent studies have also associated the abuse potential of propofol with the glucocorticoid system. Intra NAc administration of dexamethasone, a glucocorticoid receptor agonist increased the propofol addictive behavior in rats. This was further associated with upregulation of drD1 in NAc [42,43]. In another study, the Nitric oxide system was seen to play a functional role in anesthetic and locomotor stimulant effects of propofol [44,45]. Shahzadi et al. [20] showed that propofol-induced conditioned place preference (CPP), one of the behavioral test for rodents, was attenuated by inhibition of nitric oxide synthetase.

Drugs of addiction such as ethanol inhibit N-methyl-D-aspartate receptors (NMDAR) in the NAc after initial acute exposure. Inhibition of NMDA receptors is associated with activation of mammalian target of rapamycin complex 1 (mTORC1), a kinase responsible for alcohol associated synaptic plasticity, learning and memory in drD1 expressing NAc neurons. On the contrary, repeated cycles of alcohol cause long-term potentiation of NMDAR in striatal cells [46,47]. Propofol is also seen to inhibit NMDAR NR1 subunit phosphorylation at Ser897 and Ser896 through a signaling mechanism involving protein phosphatase 2A. However, this effect was observed in cortical neurons and it will be interesting to know how the NAc NMDA are affected in acute and chronic administration of propofol [48,49]. 

While assessing the function of glycine receptors in anesthesia-induced hypnosis, we observed that strychnine, a glycine receptor antagonist, reduced the LORR (loss of righting reflex), a marker of the hypnotic state induced by propofol [50]. In a study by Molander and colleagues [51], reversed microdialysis of glycine receptor agonist into the NAc increased release of NAc dopamine while infusion of glycine receptor antagonist decreased dopamine levels. Glycine receptors have also been involved in dopamine-enhancing effects of addictive drugs such as ethanol, tetrahydrocannabinol and nicotine [52,53]. Glycine receptors, thus playing a part in propofol induced hypnosis, could possibly play a part in its addiction circuitry too. On the other hand, GABA receptor has been implicated in developing synaptic plasticity to benzodiazepines, particularly α2 and α3 subunits containing GABA_A_ receptor, which cause a disinhibition of dopamine neurons as observed with other drugs of abuse [54,55]. The principle site of action of propofol being on the GABA_A_ receptor, this receptor can be another plausible substrate involved in propofol addiction mechanism too.

Recent studies explore the role of micro-RNAs in the addiction cycle of drugs [56,57]. Micro-RNAs are endogenous, small, non-coding RNAs that negatively regulate expression of multiple genes in a cell via degradation or translational inhibition of the target mRNA. Particularly, Li et al. [58] in their molecular study observed that downregulation of miRNA during alcohol dependence causes increased drD1 and ΔFosB expression observed during alcohol dependence, with ΔFosB being a downstream signal molecule for drD1. This increased expression can be inhibited by overexpression of miR382 (MicroRNA 382), in turn decreasing the voluntary intake as well as preference for ethanol. Whether propofol addiction also has a characteristic to regulate the microRNAs will be interesting to investigate in future especially in the light of similarity in terms of increased expression of drD1 and ΔFosB seen with propofol studies.

In a nutshell, despite having distinct targets, different drugs of addiction share the common reward pathway by enhancing synaptic secretion of dopamine from the VTA on to the NAc [25]. Propofol also increases the NAc dopamine levels following acute exposure and further modulates the VTA-NAc circuit following repeated exposure (Figure 1). It will be helpful to know which of the different receptor targets of propofol act as the primary substrate re-enforcing the addictive potential of propofol. Subsequently, exploring the possibility and mechanism bridging the synaptic plasticity-induced changes by repeated propofol use and the resultant addictive phenotype will be of paramount importance in determining the significance of the proposed future regulatory recommendations on propofol use.

## 6. Conclusions

There has been increasing evidence that propofol mimics the actions of other addictive substances such as alcohol and nicotine and poses a risk for addiction and mortality. Studies focusing on the subjective psychotropic effects of propofol and the rising incidence of propofol abuse point to the increasing risk of propofol as a significant cause of substance abuse, especially among healthcare professionals who have easy access to propofol. Studies concentrating on propofol’s effect on the mesolimbic system show similarities with other drugs of abuse in terms of molecular cascades that induce behavioral sensitization and dopaminergic activation. Growing evidence that propofol can be self-administered shows a conditioned place preference and produces a discriminable stimulus similar to known drugs of addiction that supports the abuse potential for propofol. This, combined with the high rate of mortality from repeated propofol use, warrants a necessity to develop regulations on its accessibility. Despite these risks, propofol is easily accessible to healthcare workers due to lack of restrictions and accountability in most healthcare settings. With growing evidence that propofol poses an increased risk of addiction and abuse, hospitals and regulation agencies should consider certifying propofol as a controlled substance to minimize incidences of morbidity and mortality from its abuse. 

## 7. Future Directions

The focus on propofol addiction appears to be in its preliminary stages. Though there has been growing conviction directing towards the potential of propofol for abuse, the addictive characteristic of propofol is still a matter of debate in the research arena. The scattered and sporadic nature of studies demand more evidence at clinical as well as experimental levels. The modes of addiction to propofol need to be explored further to determine the population at risk of its use. While these measures can help in determining the gravity of the problem and methods to exert control over its harmful consequences, detailed work at the cellular, molecular, and genetic levels as suggested in this review may prove essential to develop strategies to curtail the problem. 

## Figures and Tables

**Figure 1 brainsci-08-00036-f001:**
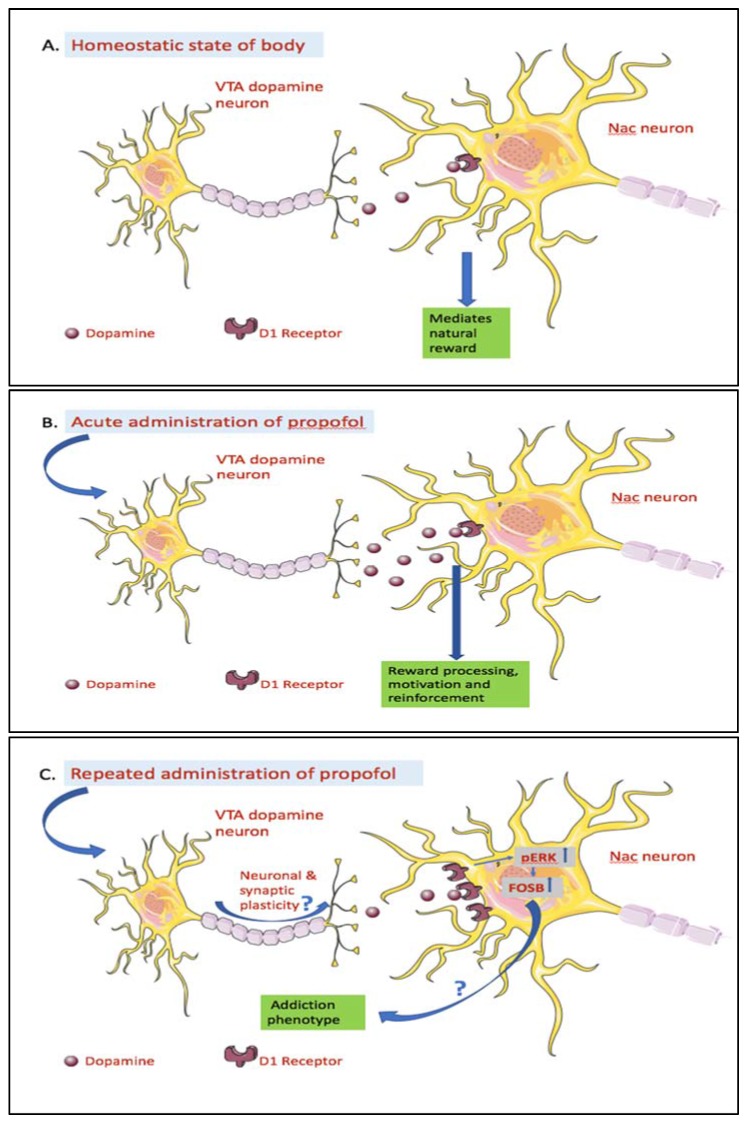
Effects of acute and repeated administration of propofol on the mesolimbic system. (**A**) Homeostatic state of body: The Ventral Tegmental Area-Nucleus Accumbens (VTA-NAc) circuit is a part of mesolimbic pathway and in homeostatic state of the body is involved in mediating individual’s responses to natural rewards like food, water, sex, and nurturing. This circuit comprises of dopamine neuron cell bodies in the Ventral Tegmantal Area (VTA) and their axons terminate in the Nucleus Accumbens (NAc) [59]. (**B**) Acute administration of propofol: Nanomolar (0.1–1 nM) concentrations of propofol enhanced VTA dopamine neuron activity by increasing the glutamatergic transmission to VTA dopamine neurons via presynaptic D1 dopamine receptors. This increased Dopaminergic (DA) neuron’s activity may cause increase dopamine levels in the NAc [34]. As such, acute intraperitoneal administration of sub-anesthetic and anesthetic doses of propofol resulted in increased dopamine levels in NAc [27]. Dopamine release in the NAc has been associated with motivation, reinforcement and reward processing [60,61]. (**C**) Repeated administration of propofol: Repeated intraperitoneal injection of propofol resulted in upregulation of dopamine receptor D1 and its downstream signaling molecules phosphorylated-EKR (p-ERK) and thereby the oncogene homolog B delta variant (ΔFosB) in the NAc [33,62]. The neuronal and synaptic plasticity in the VTA-NAc circuit, as seen with other drugs of abuse [25,26], may play a role in development of these changes. However, any experimental verification to support this theory remains to be elucidated. ΔFosB has been demonstrated to contribute to drug addiction by regulating expression of target genes in the reward pathway [31,32]. This evidence may hint towards a similar mechanism underlying propofol addiction and thus needs further research supporting this proposition.

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
