# Peer review of "Neurobiology of Propofol Addiction and Supportive Evidence: What Is the New Development?"

_brainsci, 2018, doi:10.3390/brainsci8020036_

Round 1

Reviewer 1 Report

This review article is interesting and has value, but some items need to be changed before it can be published.

  The Drug Enforcement Agency (DEA).

Generic drug names are written in lower case (e.g., propofol).  Both upper and lower case are used in the manuscript.

Widespread use for sedation in intensive care units of hospitals (ICUs).  

Mention of fospropofol under the section describing status of propofol as a controlled drug is confusing.  The authors need to note that fospropofol is no longer available in the U.S., even though it was once a CIV drug.  There is nothing in the first paragraph of section 4 that even lets the reader know that it is the prodrug of propofol.

This reviewer could not find a statement from the American Society of Anesthesiologists recommending CIV classification for propofol, although they routinely publish how to use this anesthetic safely.  The URL provided was from the Department of Justice for 5-Methoxy-N,N-Dimethyltryptamine.

The AANA recommendation on propofol is not referenced.

Section 5:  The abuse potential of propofol observed clinically can be substantiated with animal studies. 

 When citing contents of a specific reference, it would be good to identify that reference up front.  For example, Wang et al. (15), focused on the significance of extracellular signal-regulated kinase....

Be more formal:  Our laboratory studied....

Section 5 is very long and text intense.  It is good to bring in those animal studies, but even the title of this section does not let the reader know the authors' intention.

This manuscript would benefit from intensive review by clinician who uses propofol in his/her medical practice.

This manuscript would also benefit from intensive review for English and use of English grammar.

Author Response

Reviewer 3:

Thank you for your comments. Your suggestions are very helpful for the manuscript. We have made the following changes to address the issues that you have highlighted;

Comment 1: The Drug Enforcement Agency (DEA).

Response: Correction is done (Line 19, 93)

Comment 2: Generic drug names are written in lower case (e.g., propofol).  Both upper and lower case are used in the manuscript.

Response: Correction done at all places in the manuscript

Comment 3: Widespread use for sedation in intensive care units of hospitals (ICUs).  

Response: Correction is done (Line 41)

Comment 4: Mention of fospropofol under the section describing status of propofol as a controlled drug is confusing.  The authors need to note that fospropofol is no longer available in the U.S., even though it was once a CIV drug.  There is nothing in the first paragraph of section 4 that even lets the reader know that it is the prodrug of propofol.

Response:

We appreciate the author’s suggestion. Fospropofol, once popular as a water-soluble prodrug of propofol, is phased out in US market due to its unreliable bioavailability and its classification as a class IV scheduled drug by Drug Enforcement Agency since 2010. We have included a brief mention of fospropofol being a prodrug of propofol in the manuscript (Line 94) as suggested by the reviewer. Our intention of referring to fospropofol in the section 4 is to compare its mechanism of action and its abuse potential to its active metabolite, propofol, which also requires a vigilant accountability as suggested by ASA, AANA and DEA.

Comment 5: This reviewer could not find a statement from the American Society of Anesthesiologists recommending CIV classification for propofol, although they routinely publish how to use this anesthetic safely.  The URL provided was from the Department of Justice for 5-Methoxy-N, N-Dimethyltryptamine.

Response: The ASA “supports” the proposal by DEA to include propofol in the list of schedule IV of control substances. Reference added (Ref no. 15)

Comment 6: The AANA recommendation on propofol is not referenced

Response: Reference added (Ref no. 16)

Comment 7: Section 5:  The abuse potential of propofol observed clinically can be substantiated with animal studies. 

When citing contents of a specific reference, it would be good to identify that reference up front.  For example, Wang et al. (15), focused on the significance of extracellular signal-regulated kinase....

Be more formal:  Our laboratory studied....

Response: Corrections done (Lines 73, 142, 171, 178, 186, 195, 213, 226, 238)

Comment 8: Section 5 is very long and text intense.  It is good to bring in those animal studies, but even the title of this section does not let the reader know the authors' intention.

.

Response: Title to section 5 has been modified. Unlike other addictive substances, the concept of propofol addiction has met with resistance due to its unique characteristics (such as pain on injection, rapid onset time) and strong medical requirement (large usage). Animal studies on propofol addiction are still in their preliminary stage and multiple possibilities may exist around its mechanism of action. The author’s intention is to present all such possible mechanisms and solicit more research in this respect to provide convincing evidence for propofol addiction. Working on these lines may bring forth treatment modalities to curb this mortal addiction disorder.

Comment 9: This manuscript would benefit from intensive review by clinician who uses propofol in his/her medical practice

Response: The manuscript has been reviewed by several Anesthesiologists who are using propofol regularly in their practice. The “awareness about propofol addiction” seems to be more or less deficient in the practicing population and they all agree on “lack of accountability of propofol usage”. Propofol can be easily accessible from the waste bucket. Hence it deems important to spread awareness as well as consider implementation on its controlled use.

Comment 10: This manuscript would also benefit from intensive review for English and use of English grammar.

Response: The section is reviewed for the language and grammar errors and edited as per suggestions.

Ming Xiong

Anesthesiology Department

NJMS-Rutger

Reviewer 2 Report

This is a well presented review on the abuse potential of propofol, a very commonly used and unregulated substance. It is quite clear, both through literature review and through anecdotal evidence from both patients and colleagues in the medical field that this is a real problem. Propofol has been proven to cause a euphoria that can lead to abuse, as the authors outline in this review. The authors did a great job at describing mechanisms of abuse potential, and the figure presented speaks to this. 

I have a couple minor comments about this article.

1) The authors open with the high-profile case of propofol abuse by Michael Jackson; however, I would venture to say that though important to mention, this does not necessarily build the case very well. The reason being that the access in this high-profile case (having a personal physician willing to provide such illigitimate services) is thankfully not something that is easily accessible to the general public. On the other hand, there are a number of reported propofol abuse cases by people who are not health care workers that would do a better job of demonstrating that this can be a problem among "lay people." 

2) It would be very helpful if the review included a discussion about the challenges of regulating propofol. It seems that if this was something that would be easily done, it would have been already started; however, the resistance to regulating propofol is likely due to either real or perceived difficulties in controlling this substance. A discussion regarding the potential increase in documentation burden, feasibility (or lack thereof) of monitoring use of propofol, and the potential increased cost of such regulations would help illustrate why there is a resistance to regulating propofol. Essentially, a discussion about the cost-benefit of regulating propofol would help improve this review. 

Author Response

Reviewer 2:

Thank you for your comments. Your suggestions are very helpful for the manuscript. We have made the following changes to address the issues that you have highlighted;

Comment 1: The authors open with the high-profile case of propofol abuse by Michael Jackson; however, I would venture to say that though important to mention, this does not necessarily build the case very well. The reason being that the access in this high-profile case (having a personal physician willing to provide such illigitimate services) is thankfully not something that is easily accessible to the general public. On the other hand, there are a number of reported propofol abuse cases by people who are not health care workers that would do a better job of demonstrating that this can be a problem among "lay people."

Response: We fully agree with reviewer’s comments. We have added a few more references with elaboration to highlight the problem of propofol abuse in the lay people. This has been added at the opening of Section 3 (Line 55). Addiction is an epidemic disease and drug seeking behavior may be encouraged if there is “no accountability” of how much propofol is used in patients and how much is wasted after a case. The easy accessibility (eg. Re-using the propofol found in the “waste bucket”) would be a way of propofol being available outside the operating rooms to lay people.

Comment 2: It would be very helpful if the review included a discussion about the challenges of regulating propofol. It seems that if this was something that would be easily done, it would have been already started; however, the resistance to regulating propofol is likely due to either real or perceived difficulties in controlling this substance. A discussion regarding the potential increase in documentation burden, feasibility (or lack thereof) of monitoring use of propofol, and the potential increased cost of such regulations would help illustrate why there is a resistance to regulating propofol. Essentially, a discussion about the cost-benefit of regulating propofol would help improve this review. 

Response: Due to unique features of propofol, there is a large necessity of this drug in clinical and hospital settings. Strict accountability may add to the economic burden on the hospital. However, the authors believe that the benefit arising out of regulating propofol can outweigh the increase in cost. We have added a paragraph explaining the challenges to control propofol use in the end of section 4 (Line 109). As described in the manuscript, propofol addiction is very “difficult to treat” and the “associated mortality is unusually high”.

Ming Xiong

Anesthesiology Department

NJMS-Rutgers

Reviewer 3 Report

Neurobiology of propofol addiction and supportive evidence: What is the new development? by Xiong et al reviews prior literature on propofol abuse.  The manuscript provides a scientific basis for addiction. The manuscript notes the concern that propofol is not placed on the list of controlled substances and currently is only a schedule IV drug.

The argument for abuse is strong and the concern for addiction is heightened by the manuscript.

No further comments.

Author Response

Reviewer 1:

Thank you for your comments on the topic. We are delighted to know that the topic seems to be of clinical relevance and interest according to you.

Ming Xiong

Anesthesiology Department

NJMS-Rutgers